# Root remodeling mechanisms and salt tolerance trade-offs: The roles of HKT1, TMAC2, and TIP2;2 in Arabidopsis

Nouf O. Alshareef[1], Vanessa J. Melino[2,3], Noha Saber[2], Annamaria De Rosa[4], Elodie Rey[2], Jian You Wang[2], Salim AlBabili[2], Caitlin Byrt[4], Mark A. Tester[2], Magdalena M. Julkowska[2,5*]

1 Department of Biochemistry, Faculty of Science, King Abdulaziz University, Jeddah, Saudi Arabia, 2 Center for Desert Agriculture and Division of Biological and Environmental Sciences and Engineering, King Abdullah University of Science and Technology (KAUST), Thuwal, Saudi Arabia, 3 Faculty of Science, Centre for Plant Science, School of Environmental and Life Sciences, University of Newcastle, Callaghan, New South Wales, Australia, 4 Division of Plant Sciences, Research School of Biology, College of Science, Australian National University, Acton, Australia, 5 Boyce-Thompson Institute, Ithaca, New York, United States of America

☯ These authors contributed equally to this work.
* mmj55@cornell.edu

## Abstract

Plant responses to salt stress involve regulatory networks integrating ion transport, hormonal signaling, and root system architecture remodeling. A key adaptive mechanism is the regulation of sodium (Na⁺) transport by Class 1 HKT1 transporters, which compartamentalize Na⁺ in non-photosynthetic tissues. High HKT1 expression reduces Na⁺ accumulation in shoots, leading to increased salt tolerance, but simultaneously results in reduced lateral root development. In this study, we explored transcriptional responses that are altered by high HKT1 expression in root stelle in two Arabidopsis backgrounds, Col-0 and C24. We identified *TMAC2*, a negative ABA regulator, and *TIP2:2*, a tonoplast aquaporin, as key modulators of root development under salt stress. While *TIP2:2* function was conserved, *TMAC2* exhibited genotype-specific effects on ABA accumulation and *HKT1*-mediated salt sensitivity. Co-expression of *TMAC2* and *HKT1* in *Col-0* upregulated *ABI4* and *ABI5*, linking Na⁺ transport to ABA signaling. Our findings underscore genetic context in shaping salt responses and provide molecular targets for enhancing root plasticity under stress.

### Author summary

Plants face significant challenges when growing in saline soils, requiring them to balance sodium (Na⁺) exclusion with maintaining root growth. While High-Affinity Potassium Transporter 1 (HKT1) reduces Na⁺ accumulation in shoots, its expression in root stele leads to reduced lateral root development, highlighting a trade-off between salt tolerance and root plasticity. Here, we characterized

**Data availability statement:** o The RNA-sequencing data generated in this study have been deposited in NCBI SRA under the accession number PRJNA1246062. Additional datasets, including raw and processed gene expression data, phenotypic measurements, and statistical analysis scripts, are available in the supporting information files. The Arabidopsis transgenic lines used in this study (UASGAL4:HKT1, TMAC2 overexpression, and TIP2;2 mutants) are available from Arabidopsis Resource Center (https://abrc.osu.edu/) For any additional reagents, plasmids, requests can be directed to the corresponding author.

**Funding:** Funding for N.A., V.J.M, J.Y.W, S.A., M.A.T, M.M.J was provided through KAUST baseline funding for M.A.T, and BTI start-up funding for M.M.J.; C.B and A.M. were supported by the Australian Research Council (FT180100476). The funders / funding bodies played no role in in the study design, data collection and analysis, decision to publish, or preparation of the manuscript.

**Competing interests:** The authors have declared that no competing interests exist.

transcriptional responses to high stele-expression of HKT1 and salt stress. We identified two key regulators of this process: **TMAC2**, a modulator of abscisic acid (ABA) signaling, and **TIP2;2**, a tonoplast aquaporin involved in Na$^+$ compartmentalization. Our results show that TMAC2 expression is key in regulating salt-induced changes in ABA levels, that subsequently reduce lateral root development. Meanwhile, TIP2;2 facilitates vacuolar Na$^+$ sequestration, further contributing to root system remodeling under salt stress. Our findings provide new insights into how plants integrate Na$^+$ transport, hormone signaling, and root growth responses to cope with salinity. Understanding these regulatory networks may help improve crop resilience by targeting key genes involved in root adaptation to stress.

## Introduction

Salinity stress is a major environmental constraint that affects plant growth and productivity, necessitating diverse adaptive strategies to maintain water and nutrient homeostasis. While substantial research has focused on mechanisms such as sodium (Na$^+$) exclusion [1–3], ion sequestration [4,5], and early sodium-independent responses to salt stress [6–8], the role of root system architecture (RSA) in salinity tolerance remains underexplored [9,10]. Roots serve as the first point of contact with saline soils, and their ability to modify growth patterns can significantly impact a plant's ability to cope with salt stress [11]. Despite this, the molecular regulators linking RSA modifications to salinity tolerance remain largely unknown.

A key player in ion exclusion during salt stress is the High-affinity Potassium Transporter (HKT1), which regulates Na$^+$ transport at the tissue level. Class 1 HKT1 proteins primarily mediate Na$^+$ retrieval from the xylem, reducing shoot Na$^+$ accumulation and delaying premature leaf senescence [12–15]. Natural variation in HKT1 expression correlates with shoot Na$^+$ content in Arabidopsis and wheat, highlighting its role in salinity adaptation [1,16,17]. Unlike other transporters such as SOS1 or GORK, which predominantly affect main root length and overall growth under salt stress [18,19], HKT1 exhibits a distinct effect on lateral root development—an aspect of RSA that has received little attention in the context of salt stress [10].

Previous work demonstrated that stele-specific overexpression of HKT1 reduces lateral root formation under salt stress, potentially due to increased Na$^+$ accumulation in the root stele and pericycle—the tissue that gives rise to lateral roots [20–22]. This observation suggests that root system remodeling under salt stress may be influenced by Na$^+$ distribution within the root rather than a direct role of HKT1 in root development. Notably, potassium (K$^+$) supplementation alleviates the HKT1-dependent suppression of lateral root development, suggesting a regulatory interplay between Na$^+$ transport, K$^+$ availability, and RSA modulation [10].

While many cell-type specific transcriptomic studies provide valuable insight into plant development [23–25], the cell-type specific effect of ion accumulation remains underexplored. Here, we investigated how HKT1-mediated Na$^+$ transport influences

root system architecture under salt stress by modulating lateral root development at transcriptional level. To identify genes responsive to salt in a tissue-specific manner, we examined two independent UASGAL4:HKT1 overexpression lines with distinct genetic backgrounds: J2731 (C24) and E2586 (Col-0). Both lines overexpress HKT1 in root stele, with slight difference in expression domains. HKT1 expression domain in J2731 includes the stele and pericycle, while E2586 shows HKT1 overexpression exclusively in root stele. Moreover, Col-0 and C24 differ fundamentally in their response to salt stress based on their calcium signatures responses [26]. Despite these fundamental differences, both backgrounds exhibit reduced lateral root development under salt stress, with partial rescue upon K$^+$ supplementation [10]. Through transcriptomic analysis, we identified two key regulators: TWO OR MORE ABRES-CONTAINING GENE 2 (TMAC2), a negative regulator of ABA accumulation, and AtTIP2;2, a vacuolar-localized aquaporin. While TMAC2 overexpression had differing effects on ABA accumulation and root growth between Col-0 and C24, both backgrounds supported a role for ABA in modulating root system architecture under salt stress. AtTIP2;2 contributed to lateral root inhibition, suggesting a broader regulatory network beyond Na$^+$ transport. Together, our findings indicate that root system remodeling under salinity stress is linked to Na$^+$ sequestration, compartmentalization and ABA signaling, with genotype-specific variations in regulatory pathways. This study provides new insights into the molecular mechanisms underlying root plasticity in response to salt stress, offering a foundation for future work in improving salt stress resilience.

## Materials and methods

### Plant material and growth conditions

The two available lines with enhanced expression of HKT1 at its native site of expression (at the root stelar cells) were used in this study. These lines are E2586 UAS$_{GAL4}$:HKT1 in the Col-0 background and J2731 UAS$_{GAL4}$:HKT1 in the C24 background as described in [5]. Seeds of the T-DNA insertion lines were obtained from the Arabidopsis Biological Resource Centre (ABRC) (S1 Table). The promoters and coding sequences of AtTIP2;2 and TMAC2 were cloned into GreenGate vectors [27]. The promoters and coding sequences were cloned into pGGA and pGGC, respectively, with mutated BsaI sites using primers (S2 Table). Modular entry vectors were subsequently ligated with 35S/ UBQ promoters and fluorophore tags (GFP or mCherry) into the destination vector (pGZ001) [27]. All overexpression lines were generated by agrobacterium-mediated transformation of Arabidopsis thaliana Col-0, C24, E2586, or J2731. Oligos used in generating overexpression lines are listed in S2 Table. The primers to measure the gene expression in mutant lines are listed in S3 Table.

### Root phenotyping

Salt-induced changes in root architecture were quantified according to [10]. Four-day-old Arabidopsis seedlings were transferred to ½ MS square plates containing either 0 mM NaCl (control) or 75 mM NaCl (salt treatment). Roots were scanned every 2nd day until 10 days after treatment. RSA was quantified using the Smart Root [28] ImageJ plugin (version 1.53T).

### Sample collection For RNA-Seq experiment and data analysis

Four-day-old seedlings (UAS-HKT1 lines and background line) were transferred for 24 hours to ¼ MS plates supplemented with one of the following treatments: 0 mM NaCl, 75 mM NaCl, 30 mM KCl or 30 mM KCl with 75 mM NaCl. Roots from five-day-old seedlings were collected and combined from 150 to 200 seedlings for a single biological replicate and snap-frozen in liquid nitrogen. Total RNA was isolated using Trizol [29], and mRNA was enriched using Ambion Dynabeads [30]. mRNA was used for library preparation, and cDNA libraries were sequenced, producing 150 bp paired end-reads (NovaSeq6000, Novogene) with a data output average of 15 million reads per sample and Q30 of 85%. The raw data is available at NCBI under the project number PRJNA1246062.

Raw counts were normalized by TMM (edgeR package), and differentially expressed profiles were produced from three biological replicates by size factors (DESeq package). The expression of each mapped gene relative to Control conditions is listed in S4 Table. The normalized expression (tpmr) of each gene across individual samples is listed in Supplementary S5 Table. The data was inspected for reproducibility using multi-dimensional scaling, and the differential expression was calculated using the DESeq package. We used the cutoff of -2 < log2(FC) > 2 and FDR p-value of 0.05 to identify differentially expressed genes for each condition x genotype combination using 0 mM NaCl 0 mM KCl as reference condition. We subsequently examined DEG shared between each genotype and condition and identified DEGs in response to treatment (75 mM NaCl, 30 mM KCl or 30 mM KCl and 75 mM NaCl) relative to control conditions (0 mM KCl and 0 mM NaCl), and HKT1 overexpression (E2586 vs E2586 UAS$_{GAL4}$:HKT1 and J2731 vs J2731 UAS$_{GAL4}$:HKT1). The genes showing response to both treatment and HKT1 were subsequently compared across individual background lines (Col-0 and C24) and salt stress treatments (75 mM NaCl and 30 mM KCl + 75 mM NaCl). The overlapping double-DEGs were inspected in further detail.

## Expression analysis of transgenic lines

To determine the expression level of different transgenic lines, total RNA was extracted from the leaves of soil-grown Arabidopsis plants using Direct-zol RNA MiniPrep Plus (Zymo Research, CA, USA) following manufacturer's instructions and quantified using a NanoDrop spectrophotometer. Total RNA (1 µg) was used to synthesize cDNA using iScript™ Reverse Transcription Supermix kit (Bio Rad) according to the manufacturer's instructions. qPCR was performed using SsoAdvanced Universal SYBR Green Supermix (Bio-Rad, Hercules, CA, USA; 172–5270) on CFX96 real-time PCR machine (Bio-Rad) following standard protocol. The expression level of the target gene was calculated using the dCt method and normalized to the geometric mean of actin2 (AT3G18780) and EF1a (AT5G60390). Primers for qPCR are listed in S3 Table. Gene expression was calculated for three independent biological replicates and two technical replicates.

## Subcellular localization using transient co-expression assay

To determine the subcellular localization of AtTIP2;2 and AtTMAC2, each protein was transiently expressed in *Nicotiana benthamiana* leaf epidermal cells. Co-localization of AtTIP2;2 and AtTMAC2 with a marker protein was determined using signal correlation analysis based on confocal microscopy images. Three marker proteins were used in this study; plasma membrane protein 1;4 (PIP1;4) as a plasma membrane marker, AtVAMP711a as a vacuolar marker [31], and Serrate as a nuclear marker, being expressed specifically in the nucleoplasm [32].

To obtain the expression vectors for protein transient expression, coding sequences of AtTIP2;2, AtTMAC2, AtPIP1;4, and VAMP711a were amplified from the cDNA of Col-0 Arabidopsis plants using Phusion High-Fidelity DNA Polymerase (New England Biolabs) and primers described in (S2 Table). Amplicons were purified and cloned into GreenGate entry vector pGGC (Addgene plasmid #48858; http://n2t.net/addgene:48858) to generate entry vectors (without stop codon). The entry vectors were subsequently cloned into the GreenGate [27] expression destination vector pGGZ001 (Addgene plasmid #48868; http://n2t.net/addgene:48868) using BsaI digestion and T4 DNA ligation (respectively M0202S and M0202S both from NEB). Correct ligation was confirmed by digestion with enzymes XmnI, BbsI, and BsmAI from NEB.

The 35S::AtTIP2;2-eGFP, 35S::AtTMAC2-eGFP, 35S:mcherry-AtPIP1;4, 35S::mCherry-VAMP711a, 35S::CFP-SERAT constructs were transformed into Agrobacterium tumefaciens strain GV3101 [33] using a heat-shock transformation protocol [34]. Leaves from four-week-old *N. benthamiana* were inoculated with agrobacterium culture as follows: protein of interest (AtTIP2;2 or AtTMAC2), marker (AtPIP1;4 or AtVAMP711) and P19 (viral gene silencing suppressor to enhance transient expression efficiency [35]. Co-localization was visualized 72 hr post-infiltration. Images were captured using a confocal laser scanning microscope (Stellaris, Leica). Images were analyzed using LAS X software (Leica). The eGFP excitation was at 488 nm, and the emission was between 500 and 530 nm, while the excitation for mCherry was at 561 nm, and the emission was at 580–630 nm, and for CFP 405 nm as excitation and 432–513 nm for emission.

## Localization of TIP2;2 and TMAC2 in stable transformants

To confirm localization in Arabidopsis under control of the native promoter, Arabidopsis transformants harboring either AtTIP2;2::AtTIP2;2-eGFP or AtTMAC2::TMAC2-eGFP were generated. Leaves and roots of ten days old Arabidopsis seedlings grown in ½ MS plates were harvested and visualized using confocal microscopy. Roots were stained with propidium iodide PI (10 mg.ml-1) for 1 min to better visualize the cellular structures of root cells. The PI was excited at 568 nm, and the emission was 585–610 nm. Images were captured using a confocal laser scanning microscope (Stellaris, Leica).

## Measurement of Na$^+$ And K$^+$ content

Shoots and roots of Arabidopsis plants were collected after 21 days from transfer treatment plates (either 0 mM NaCl or 75 mM NaCl). Samples were then oven-dried at 75 ºC for about 3 days, digested by adding 5 mL of freshly prepared 1% (v/v) nitric acid (Sigma Aldrich), and incubated at 60°C for 2 days. The concentrations of sodium and potassium ions in the samples were determined using a flame photometer (model 425, Sherwood Scientific Ltd., UK), (Awlia 2019, Flame Photometry Protocol. https://dx.doi.org/10.17504/protocols.io.6t6here).

## ABA quantification

Quantification of endogenous hormones was performed following the procedure in [36]. Up to 15 mg of freeze-dried ground root or shoot base tissues were spiked with internal standards D6-ABA (10 ng) and 1700 μL of methanol. The mixture was sonicated for 20 min in an ultrasonic bath (Branson 3510 ultrasonic bath), followed by centrifugation for 5 min at 14,000 × g at 4°C. The supernatant was collected and dried under a vacuum. The sample was re-dissolved in 120 μL of acetonitrile:water (25:75, v-v) and filtered through a 0.22 μm filter for LC–MS analysis.

ABA was analyzed by LC-MS/MS using UHPLC-Triple-Stage Quadrupole Mass Spectrometer (Thermo Scientific$^{TM}$ Altis$^{TM}$). Chromatographic separation was achieved on the Hypersil GOLD C18 Selectivity HPLC Columns (150 × 4.6 mm; 3 μm; Thermo Scientific$^{TM}$) with mobile phases consisting of water (A) and acetonitrile (B), both containing 0.1% formic acid, and the following linear gradient (flow rate, 0.5 mL/min): 0–10 min, 15%–100% B, followed by washing with 100% B for 5 min and equilibration with 15% B for 2 min. The injection volume was 10 μL, and the column temperature was main-tained at 35°C for each run. The MS parameters of Thermo Scientific$^{TM}$ Altis$^{TM}$ were as follows: negative mode, ion source of H-ESI, ion spray voltage of 3000 V, sheath gas of 40 arbitrary units, aux gas of 15 arbitrary units, sweep gas of 0 arbitrary units, ion transfer tube gas temperature of 350°C, vaporizer temperature of 350°C, collision energy of 20 eV, CID gas of 2 mTorr, and full width at half maximum (FWHM) 0.4 Da of Q1/Q3 mass. The characteristic Multiple Reaction Monitoring (MRM) transitions (precursor ion→product ion) were The characteristic MRM transitions (precursor ion→prod-uct ion) were 263.2→153.1, 263.3→204.1, 263.3→219.1 for ABA; 269.2→159.1 for D6-ABA.

## Screening for putative solute transport with high-throughput yeast-based assays

High throughput yeast microculture screens were used to test for putative transport of diverse key plant solutes (sodium, water, $H_2O_2$, boron and urea), associating growth enhancements or toxicity phenotypes to expression of foreign aquaporins (AQPs) in yeast cells in response to various treatments.

AtTIP2;1, AtTIP2;2 and AtTIP2;3 coding sequences with stop codon were cloned into Gateway-enabled expression vector pRS423-GPD-ccdB-ECFP [37] containing Uracil3 (URA3) yeast selection gene. Yeast expression vectors were transformed in respective yeast strains required for functional assays (described below), using the "Frozen-EZ yeast Trans-formation Kit II" (Zymo Research, Los Angeles, USA). Transformed colonies were grown in Yeast Nitrogen Base, YNB, media (Standard drop out, DO, -URA) and spotted (10μl) on agar YNB (DO -URA) selection plates for incubation at 30°C for 2 days, then stored at 4°C. Spotted plates (1/2 spot per tube) were used for the starting cultures of functional assays.

Water, hydrogen peroxide (H$_2$O$_2$), boron and urea functional assays were conducted as previously described in [38]. Yeast lacking native aquaporin isoforms *aqy1aqy2* (null aqy1 aqy2; background Σ1278b; genotype: Mat α; leu2::hisG; trp1::hisG, his3::hisG; ura352 aqy1D::KanMX aqy2D::KanMX, [39] was used for the water/Freeze-thaw assay. The freeze-thaw assay exploits a property of yeast cells where increased freezing tolerance is conferred when the yeast is expressing functional water-permeable AQPs relative to yeast lacking functional water-permeable AQPs. The *aqy1aqy2* yeast was also used for a Boric Acid (BA) toxicity assay, which involved screening for enhanced AQP-associated sensitivity of yeast cells upon exposure to increasing BA treatment concentrations. H$_2$O$_2$ permeability was assessed using a reactive oxygen species (ROS) hypersensitive yeast strain, Δskn7 (null skn7; background BY4741 genotype: Mat α; his3Δ1 leu2Δ0 met15Δ0 ura3Δ0 ΔSKN7) [40,41]. In the ROS sensitivity assay yeast experienced enhanced toxicity responses if they were expressing AQPs that facilitated diffusion and accumulation of H$_2$O$_2$ in the cell. The Urea growth-based assay involved using *ynvwI* yeast (null dur3; background Σ23346c; genotype: Mat α, Δura3, Δdur3) containing a deletion of the DUR3 urea transporter [42]. In the Urea assay expression of urea-permeable AQPs in *ynvwI* yeast provided a growth advantage when exposed to media containing urea as the sole nitrogen source.

Our assay methodology framework was further adapted to establish a novel sodium toxicity screen using the salt sensitive yeast strain AXT3 (Δena1–4::HIS3 Δnha1::LEU2 Δnhx1::TRP1 [43]. AXT3 yeast cell survival was further compromised if they were expressing AtTIPs that facilitated the diffusion and accumulation of Na+ in the yeast cell, because this enhanced the toxicity response of the cell upon exposure to increasing NaCl treatments. AXT3 yeast expressing AtTIP2 isoforms and Empty vector control yeast were grown for 24 hours in 1 mL YNB(-URA) 10mM ethylene glycol-bis (β-aminoethyl ether)-N,N,N′,N′-tetraacetic acid (EGTA) pH6 media, at 30°C, shaking at 250rpm and diluted to 0.6x107 cells/mL. 200μL microcultures of each AtTIP2/Empty vector were distributed in 96-well plates with 180μL of yeast and 200μL NaCl treatments: 0mM/water, 50mM, 100mM and 200mM NaCl.

Yeast microculture growth for all solute transport assays was monitored in 96-well plate using a SPECTROStar nano absorbance microplate reader (BMG Labtech, Germany) at 10–20-minute intervals over 42–60 hours. Data collection and processing for all solutes screened was conducted as described in [38]. Briefly, growth curves were integrated using the natural log of OD$_{650}$/initial OD$_{650}$ (Ln(OD$_t$/OD$_l$) vs time) up to a time when the growth rate of the Untreated culture had declined to 5% of its maximum. Area Under the Curve (AUC), was calculated as a proxy that captured the potential growth characteristics affected, regardless of the treatment, in a single parameter. Statistical analyses were conducted using GraphPad PRISM Version 9 (San Diego, California, USA, www.graphpad.com). Error bars on graphs represent SEM. One-way ANOVAs and Post-Hoc Fisher's LSD tests were conducted to test statistical difference of means between each AtTIP and Empty vector control. Asterisks were used to denote statistical significance *p < 0.05, **p < 0.01 and ****p < 0.001. For each assay, 6 biological replicates were tested over 3 experimental runs.

## Results

### Identification Of *HKT1*-dependent patterns underlying reduced lateral root development

The central hypothesis of this study is that HKT1-mediated Na+ transport affects root architecture under salt stress by modulating transcriptional networks involved in hormone signaling and lateral root development. To investigate this, we analyzed UASGAL4:HKT1 enhancer trap lines in two Arabidopsis backgrounds, Col-0 and C24, both of which exhibit HKT1-dependent reductions in lateral root development under salt stress [10]. While both lines show increased HKT1 expression in stelle, they differ in the spatial expression of HKT1, with J2731 (C24 background) including the pericycle in the expression domain, while E2586 (Col-0 background) excludes the pericycle [5]. To explore the transcriptional changes associated with HKT1 expression and salt stress, four-day-old seedlings were transferred to control (0 mM NaCl, 30 mM KCl) or salt-treated conditions (75 mM NaCl, 75 mM NaCl + 30 mM KCl) for 24 hours, after which root tissues were collected for RNA isolation and sequencing. Multidimensional scaling analysis (Fig 1A) revealed that the strongest transcriptional variations were driven by salt treatment (Dimension 1) and genetic background (Dimension 2). Specifically, samples

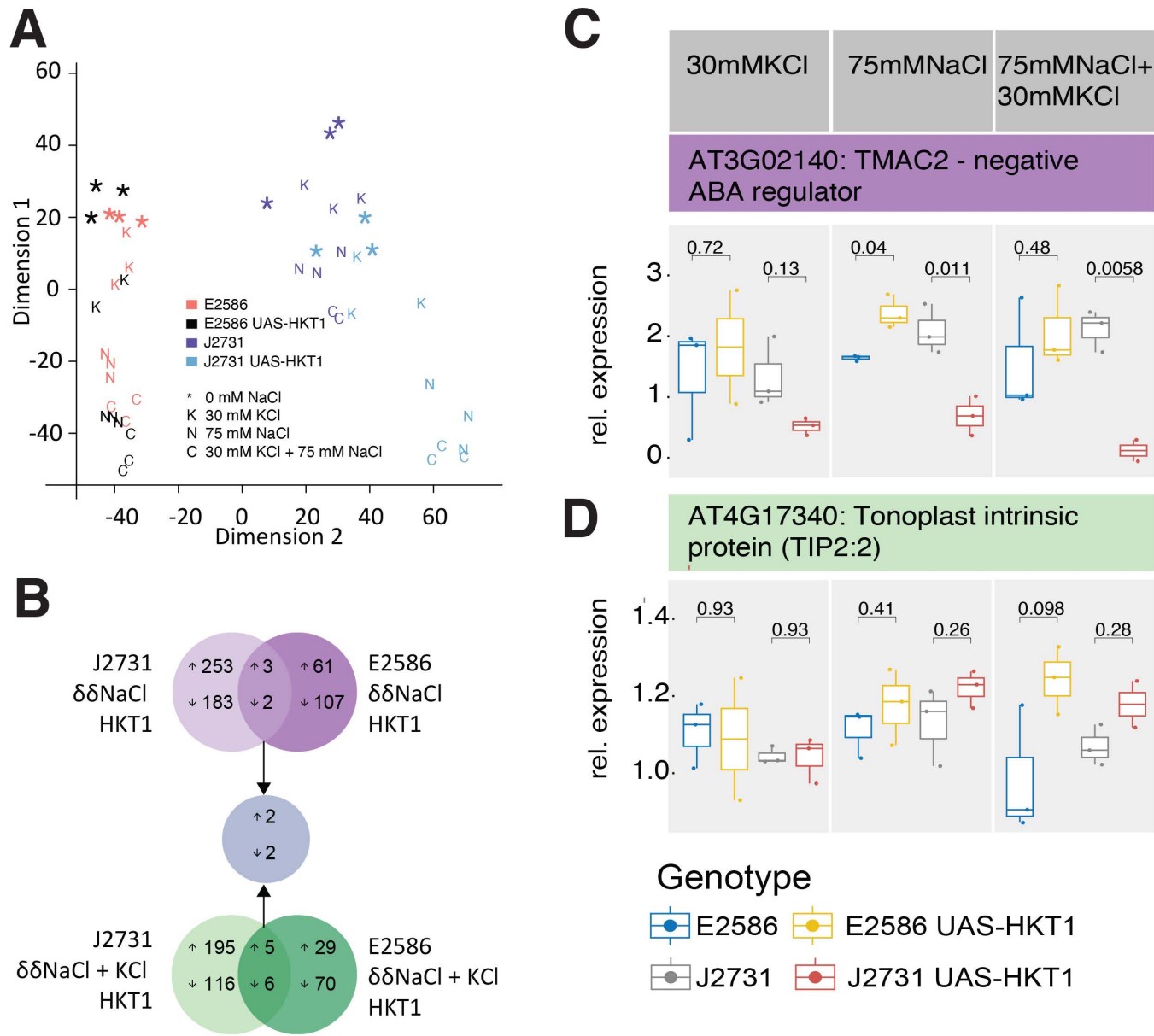

Fig 1. Identification of HKT1 and salt-induced changes on gene expression in lines with tissue-specific overexpression of HKT1. (A) Principal component analysis of variance for each RNAseq sample grouped per treatment (*, 0 mM NaCl; K, 30 mM KCl; N, 75 mM NaCl and C, 30 mM KCl + 75 mM NaCl) and per genetic line of either the controls in Col-0 (E2586) or C24 (J2731) or with $UAS_{GAL4}$:HKT1 enhancer trap lines. (B) The differentially expressed genes were grouped based on their up- or down-regulation in either treatments (δ NaCl for salt induced changes for 100 mM vs 0 mM NaCl, δ NaCl + KCl for salt induced changes for 100 mM NaCl + 30 mM KCl vs 30 mM KCl) or genetic backgrounds (δ HKT1 J2731 for UAS-HKT1 in J2731 vs J2731; δ HKT1 E2586 for UAS-HKT1 in E2586 vs E2586). The overexpression lines (UAS-HKT1) in both C24 (J2731) and Col-0 (E2586) backgrounds were compared to their respective backgrounds in response to NaCl (purple circles) and NaCl + KCl treatments (green circles). The genes shared between the genetic backgrounds and treatments were further inspected for their expression relative to control conditions across treatments, with (C) TMAC2 showing significant change in NaCl, (D) TIP2:2 showing increase in response to NaCl + KCl treatment. The p-values listed above the individual comparisons between UAS-HKT1 and their background were calculated using T-test.

from salt-treated plants clustered separately from control-treated plants, and Col-0 and C24 backgrounds exhibited distinct transcriptomic responses, reflecting both a conserved response to salt stress and genotype-specific differences (S1 Fig).

Given the strong HKT1-dependent inhibition of lateral root development, we sought to identify genes that were differentially expressed in both backgrounds in response to HKT1 overexpression and salt stress. We identified five core DEGs that were consistently altered in both backgrounds (Fig 1B, purple Venn), including three upregulated genes (AT3G02140/TMAC2, AT5G05960, AT3G18280) and two downregulated genes (AT3G11340/UGT76B1, AT2G37060/NF-YB8). AtTMAC2 (AT3G02140) emerged as a key candidate for regulating lateral root development under salt stress. This gene, encoding an ABI5-interacting protein involved in ABA signaling [44,45], was significantly upregulated in the Col-0 UASGAL4:HKT1 line under salt stress (Fig 1C). Interestingly, the opposite expression pattern in the C24 background, where salt stress alone increased TMAC2 expression, while HKT1 overexpression dampened this response. Given that TMAC2 was previously identified as a negative regulator of ABA accumulation and salinity responses [46], this suggests that HKT1 may influence lateral root development through an ABA-related feedback mechanism, but the direction of this effect is genotype-dependent.

We also identified two DEGs specifically responsive to both salt stress and HKT1 overexpression (Fig 1B, blue Venn): UGT76B1 (At3g11340), a UDP-dependent glycosyl transferase predicted to function in salicylic acid metabolism, and NF-YB8 (At2g37060), a transcription factor of the nuclear factor Y complex. Notably, no significant changes were observed in NF-YA or NF-YC components, suggesting that NF-YB8 may function independently or in a non-canonical transcriptional complex under these conditions. Two lipid transfer protein genes (At5g05960 and At3g18280) were significantly upregulated exclusively in UASGAL4:HKT1 lines under salt stress (S2 Fig). These genes belong to the bifunctional inhibitor/lipid-transfer protein 2S albumin superfamily, which has been linked to systemic acquired resistance and the deposition of cuticular waxes and suberin [47]. Their upregulation suggests that HKT1 overexpression may alter root structural integrity or defense responses under salt stress, which could influence lateral root emergence.

## Functional validation of candidate genes in lateral root development

To assess the functional relevance of the identified DEGs, we examined the phenotypes of available loss-of-function and gain-of-function mutants for several identified genes, including PHT5:1 (AT1G63010), UGT76B1 (AT3G11340), and SBT5.2 (AT1G20160) (S3, S4 and S5 Figs). Most of the genes did now exhibit salt stress induced changes in root system architecture. PHT5:1 loss-of-function mutants exhibited increased lateral root emergence under salt stress compared to Col-0 (S3A and S3B Fig), whereas gain-of-function mutants showed a significant reduction in lateral root formation (S3C Fig). UGT76B1 loss- and gain-of-function mutants did not show significant differences in lateral root development compared to Col-0 (S4 Fig), but main root length was altered under both control and salt stress conditions. SBT5.2 mutants did not exhibit lateral root changes but showed increased main root length under both control and salt stress conditions (S5 Fig). Together, these results indicate that genes differentially regulated by HKT1 and salt stress contribute to root growth and developmental plasticity, but their specific roles in lateral root emergence may vary. Given the distinct expression patterns of HKT1 in J2731 and E2586 lines, the observed differences in transcriptomic responses and lateral root phenotypes may reflect $Na^+$ accumulation patterns in the pericycle and its impact on hormone signaling.

## The effect of TMAC2 on ABA accumulation depends on genetic background and HKT1 transcription

Previous reports of *AtTMAC2* (AT3G02140) demonstrated that although roots of the RNAi mutant (*tmac2)* were similar to that of the wildtype plant, overexpression of *AtTMAC2* (*35S::TMAC2*) resulted in shorter main roots, which were insensitive to ABA-mediated inhibition [46]. The root phenotypic response of *35S::TMAC2* to salt stimuli was not described, nor were the effects on lateral root outgrowth with or without stimuli. Therefore, we explored the impact of TMAC2 on root

architecture based on its significant alteration in expression in response to both salt and HKT1 (Fig 1C). Interestingly, the expression was altered in two opposite directions in both backgrounds. TMAC2 expression was enhanced by HKT1 in Col-0 background line, while in C24 background the expression of TMAC2 was repressed in the presence of high HKT1 expression (Fig 1C). No substantial allelic variation was observed in the TMAC2 promoter region between Col-0 and C24 accessions (S6 Fig). As ABA is also known to inhibit lateral root outgrowth in response to salt stimuli mediated by the auxin-independent pathway [48], AtTMAC2 was a promising candidate for the salt-induced lateral root outgrowth inhibition observed here. TMAC2 was previously reported to be a negative regulator of ABA responses [46]. We have initially observed that the ABA accumulation pattern that reflected the transcriptional changes (Fig 2A). These results suggest that increased HKT1 expression affects TMAC2 expression differently depending on the wider genetic context, which is reflected by different ABA levels.

To test whether lateral root development can be indeed altered through increased TMAC2 expression, we have generated constitutive overexpression lines of TMAC2 (*35S::AtTMAC2*, S7 Fig) in four genetic backgrounds (Col-0 background with or without $UAS_{GAL4}$:*HKT1*, and C24 background with and without $UAS_{GAL4}$:*HKT1*). Although we anticipated that over-expression of TMAC2 would reduce lateral root development, we have only observed reduction in main root length and lateral root number in the Col-0 background (Fig 2B) with no observable change in average lateral root length (Fig 2B). No further decrease in main or lateral root number or length was observed either in C24 or in Col-0 $UAS_{GAL4}$:*HKT1* (Fig 2B). This could be caused by the fact that Col-0 $UAS_{GAL4}$:*HKT1*, C24, and C24 $UAS_{GAL4}$:*HKT1* lines were already severely impaired in lateral root development under salt stress conditions. These results suggest that increased TMAC2 expression is reducing the lateral root development exclusively in the Col-0 background in the scenario when HKT1 is not highly expressed.

In addition to the root phenotypes, we also measured ABA content in the roots and shoots of plate-grown plants (Figs 2C and S8). Surprisingly, overexpression of *AtTMAC2* in Col-0 (35S::AtTMAC2/E) led to an increased ABA content in the roots (Fig 2C), but this increase was not observed in Col-0 $UAS_{GAL4}$:*HKT1* line (Fig 2C). No difference in ABA accumulation was observed in the shoot tissue of lines overexpressing TMAC2 (S8 Fig). In contrast, there was a trend of decreased ABA content in the C24 backgrounds with overexpression of HKT1 or TMAC2 (Fig 2C). TMAC2 overexpression was not observed to have any detectable effect on sodium accumulation in both root and shoot tissues (S9 Fig).These results suggest that HKT1 reduces ABA accumulation only in the Col-0 background, while no effect was observed in the C24 background, which contradicts our initial observations (Fig 2A). On the other hand, TMAC2 overexpression by itself resulted in different ABA accumulation depending on the background. In the Col-0 background, the overexpression of TMAC2 resulted in increased ABA accumulation but not in the presence of high HKT1 expression. In C24 overexpression of TMAC2 resulted in reduced ABA levels only in combination with HKT1. These results suggest that the role of TMAC2 in regulating ABA depends not only on the genetic background but also on the levels of HKT1 expression.

As TMAC2 was confirmed to localize into the cell nucleus (Fig 3A), and the root phenotypes related to TMAC2 overexpression were only evident in the Col-0 background line, we hypothesized the existence of feedback regulation between TMAC2 and HKT1. We observed that HKT1 expression was enhanced in TMAC2 overexpression lines, even beyond the overexpression levels observed in Col-0 UAS_{GAL4}::HKT1 background line (Fig 3B). Moreover, the co-expression of both TMAC2 and HKT1 resulted in upregulation of transcription factors ABA Insensitive 4 (ABI4) and ABI5 (Fig 3C). In contrast, overexpression of TMAC2 alone did not enhance the expression of either ABI4 or ABI5 (Fig 3C). These results suggest that there is transcriptional feedback between HKT1 and TMAC2.

## TIP2;2 regulates lateral root development in response to salt independent of HKT1

The RNAseq results revealed enhanced expression of TIP2;2 in HKT1 overexpression lines in both the Col-0 and C24 backgrounds, relative to wild type lines (Fig 1D). TIP2;2 is an isoform in the tonoplast intrinsic protein subfamily of aquaporins, which are channels that can facilitate vacuolar membrane transport of water and other solutes [49–51]. To

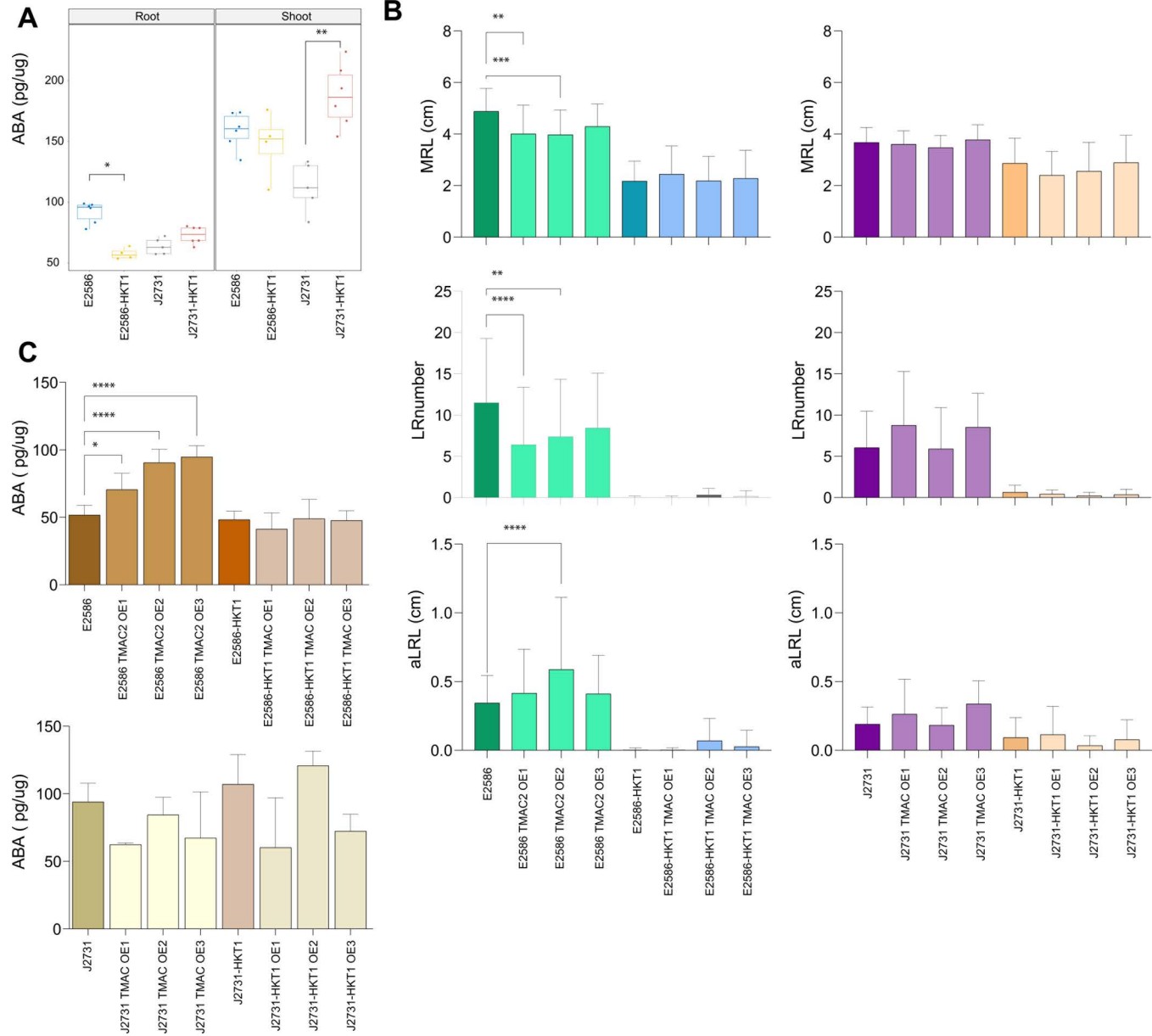

**Fig 2. TMAC2 affects ABA accumulation and lateral root outgrowth. (A)** The ABA accumulation in Col-0 (E2586) and C24 (J2731) seedlings with and without tissue specific overexpression of HKT1 was observed in 14 days old Arabidopsis seedlings, exposed to salt stress for 10 days. **(B)** Root system architecture, dissected into main root length (MRL), lateral root number (LR) and average lateral root length (aLRL) was determined for individual gain-of-function TMAC2 lines compared to their respective background linens of 13 days old plants exposed to salt stress (75 mM NaCl) for 9 days. **(C)** The ABA accumulation in the root tissue of gain-of-function TMAC2 lines and their background lines was measured from 18 days old seedlings, exposed to salt stress for 14 days. All of the measurements represent the average +/- standard error of at least 20 individual plants. The significant differences between individual mutant lines and their respective background lines were determined using one-way ANOVA test, with *, **, *** and **** indicating p-values below 0.05, 0.01, 0.001 and 0.0001 respectively.

investigate the function of the Arabidopsis TIP2;2 we evaluated the TIP2;2 T-DNA mutant in Col-0 background, as well as generated constitutive overexpression lines in Col-0 and C24 background with and without HKT1 overexpression (Figs 4 and S10). The TIP2;2 loss-of-function mutant showed enhanced lateral root elongation (Fig 4A), whereas the

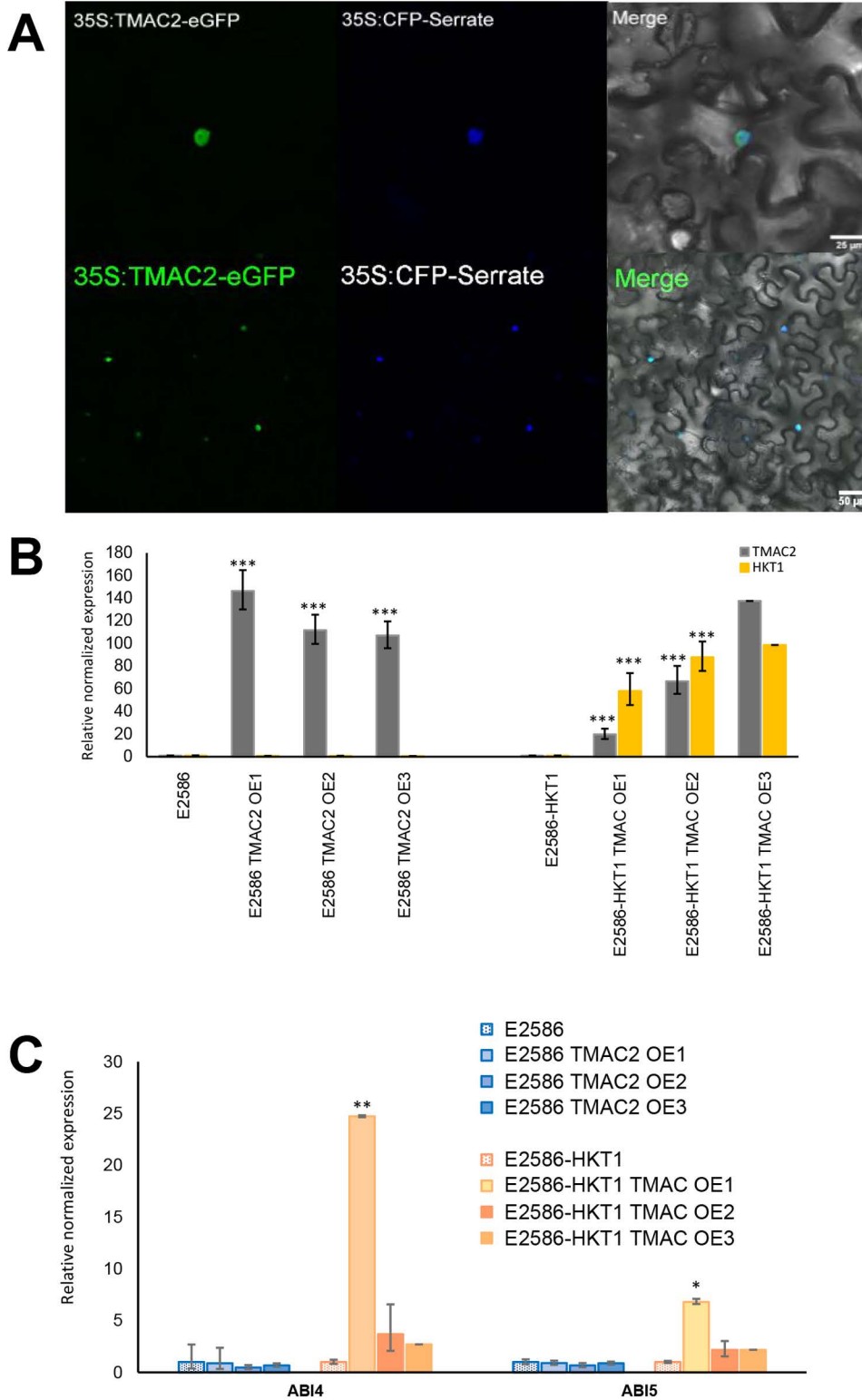

**Fig 3. Feed-forward transcriptional regulation of TMAC2 and HKT1 through ABI4. (A)** The subcellular localization of TMAC2 in transiently trans-formed *N. benthamiana* epidermal leaf cells. The cells were infiltrated with both 35S::TMAC2 construct and cell nucleus marker 35S::CFP-Serrate. The merged images illustrate co-localization of both GFP and CFP signals. **(B)** The abundance of both HKT1 and TMAC2 transcripts, **(C)** ABI4 and ABI5

transcripts relative to three housekeeping genes (Actin2, Ef-a and AtCACS (AT5G46630), measured in 18 days old seedlings of A. thaliana exposed to salt for 14 days. The graphs represent the average relative expression calculated from three biological replicates, while error bars represent standard error. The significant differences between individual mutant lines and their respective background lines were determined using one-way ANOVA test, with *, **, *** and **** indicating p-values below 0.05, 0.01, 0.001 and 0.0001 respectively.

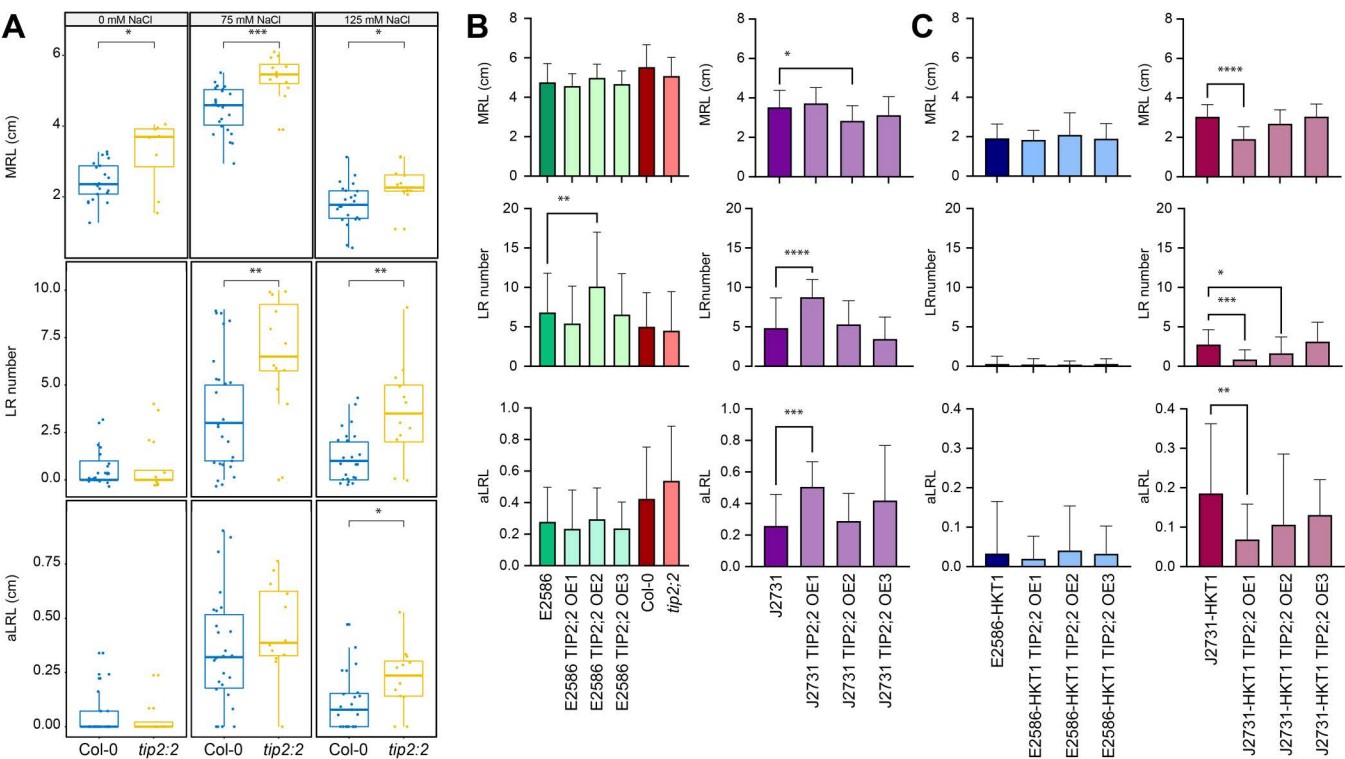

**Fig 4. TIP2;2 is a negative regulator of lateral root development under salt stress conditions. (A)** The T-DNA insertion line (*tip2;2*) was examined alongside Col-0 (wt) for salt-induced changes in root architecture. The root architecture was obtained from 8 days old seedling under non-stress conditions (0 mM NaCl) and 12 days old seedlings under salt stress conditions (75 and 125 mM NaCl). The seedlings were exposed to salt stress for 8 days. The box plots represent the interquartile range, while the thick line represents the trait mean. **(B)** The generated overexpression lines in E2586 and J2731 backgrounds, and **(C)** their respective tissue-specific HKT1 overexpression lines were examined for salt-induced changes in root architecture. The root architecture in this experiment was quantified of 13 days old seedlings exposed to 75 mM NaCl for 9 days. The bars represent the sample average and error bars represent standard error. The significant differences between individual mutant lines and their respective background lines were determined using one-way ANOVA test, with *, **, *** and **** indicating p-values below 0.05, 0.01, 0.001 and 0.0001 respectively.

overexpression of TIP2;2 in either Col-0 or C24 background did not result in noticeable differences in root architecture under control and salt stress conditions (Fig 4B), relative to wild type lines. Interestingly, overexpressing TIP2;2 in lines with tissue-specific overexpression of HKT1 did result in a further decrease of lateral root length (Fig 4C), yet the effect was only significant in the C24 background. These results suggest that TIP2;2 has a negative effect on lateral root development under salt stress and that this effect is exacerbated by high HKT1 expression.

To better understand the effect of TIP2;2 on lateral root development, we generated a translational fusion of AtTIP2;2 with GFP driven by the TIP2;2 native promoter and assessed it for sub-cellular localization (Fig 5). We observed TIP2;2 to be highly expressed in the cortex of the elongation zone of the main root and mature lateral roots (Fig 5A-H) and Arabidopsis leaf epidermal pavement cells (Fig 5I). No TIP2;2 expression was observed in the lateral root primordia or early developing lateral roots (Fig 5B, 5C, 5E, and 5H), suggesting that its role is restricted to maturing tissues. TIP2;2 was observed in structures reminiscent of the plasma membrane, vacuole, and small vesicles (Fig 5J-P). This sub-cellular

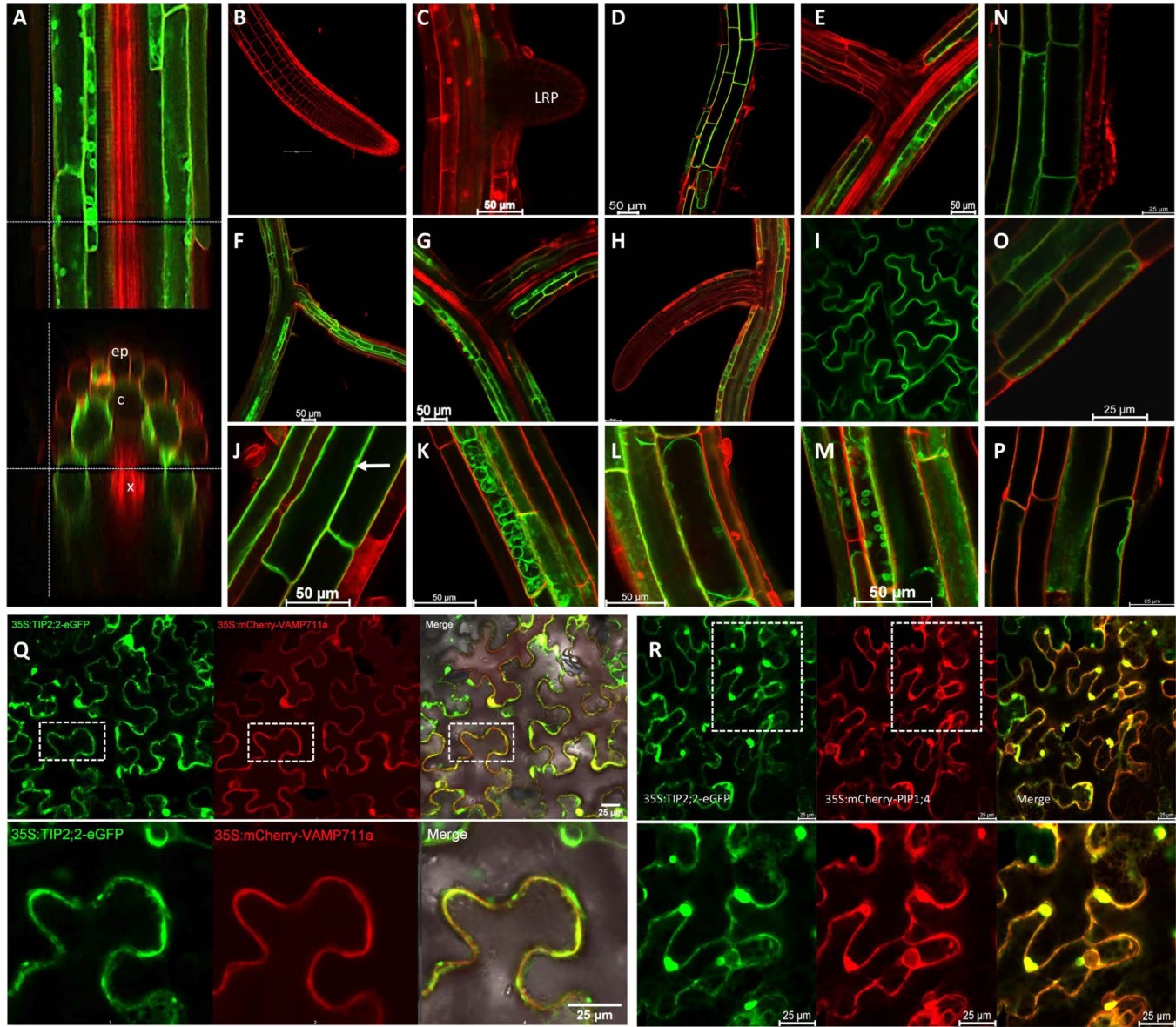

**Fig 5. TIP2;2 is expressed in the cortex layer from the elongation zone onward and localizes to both tonoplast and plasma membranes.**
Ten days old Arabidopsis seedlings transformed with pTIP2;2::TIP2;2::GFP construct were additionally stained with propidium iodide and explored for tissue-specific expression and sub-cellular localization of TIP2;2. **(A)** The optical cross-section of the root elongation zone with epidermis (ep), cortex (c) and xylem (x) indicated on the image. Further the expression was investigated in **(B)** Main Root tip, **(C)** newly developing lateral root primordia (LRP), **(D)** maturation zone of the main root, **(E)** base of the developing lateral root, and **(F-H)** fully developed lateral root, **(I)** leaf epidermal pavement cells. We additionally observed various sub-cellular localization of TIP2;2 pointing towards **(J)** plasma membrane localization, **(K)** prevacuolar vesicle localization and **(L)** tonoplast localization, and **(M)** smaller vesicle localization. The additional staining performed with the membrane dye (FM4-64) was performed for **(N-O)** 5 minutes to confirm plasma membrane localization, and **(P)** 45 minutes to confirm vacuolar localization. The transient co-expression of p35s::TIP2;2::GFP in *N. benthamiana* epidermal cells **(Q)** in combination with vacuolar marker p35S::VAMP11a::mCherry and **(R)** in combination with plasma membrane marker p35S::PIP1;4::mCherry showed co-localization of TIP2;2 with both of the markers.

localization was further confirmed through transient expression of 35S::AtTIP2;2::GFP in tobacco-infiltrated leaves (Fig 5Q-R), demonstrating that AtTIP2;2 can localize to both the vacuolar and the plasma membranes. These results confirm the tonoplast localization of TIP2;2 and identify maturing roots and leaves as the primary tissues affected by TIP2;2 expression.

To investigate TIP2;2 permeability we progressed heterologous yeast-based microculture assays. Two AtTIP2 homologs from within the TIP subfamily, AtTIP2;1 (At3g16240) and AtTIP2;3 (At5g4750) were included in the assays exploring AtTIP2;2 permeability (Fig 6). High-throughput growth- and toxicity-based yeast assays described in [38] were used to check whether these TIPs were permeable to water, urea, hydrogen peroxide and boric acid. This approach was further adapted to develop a novel sodium toxicity screen to check for putative sodium transport in TIP-expressing yeast cells, using a salt-sensitive yeast mutant strain AXT3 [43]. Yeast growth curves, Ln(OD/ODi) vs. time (S13 Fig) and Relative Area under the curve (AUC) measurements (Fig 6) revealed that NaCl sensitive AXT3 yeast expressing AtTIP2;2-exhibited reduced growth compared to Empty vector control yeast upon exposure to 50 mM, 100 mM and 200 mM NaCl treatments, indicating that TIP2;2 is a candidate for facilitating Na+ transport into the yeast cells and causing a toxicity response. AtTIP2;2 was also observed to transport water, urea, hydrogen peroxide, and boric acid (Figs 6B-E and S13). AtTIP2;1 and AtTIP2;2 were also observed to transport sodium, urea, and boric acid (Fig 6), and TIP2;3 is was not identified as candidate for water and hydrogen peroxide transport, unlike the other two TIP2 isoforms. These results reveal that there is variation in the permeability of TIP2s to different plant solutes, and indicate that AtTIP2;2 and AtTIP2;1 have similar solute transport profiles, whereas AtTIP2;3 differed.

To evaluate the function of TIP2;2 *in planta* we tested ion accumulation in TIP2;2 loss-of-function, overexpression and control lines. Overexpression of TIP2;2 did not result in enhanced sodium accumulation by itself but resulted in enhanced root $Na^+$ content in the Col-0 line that had high HKT1 expression (Fig 7A). We observed increased shoot $K^+$ accumulation in TIP2;2 overexpression lines in Col-0 $UAS_{GAL4}$:HKT1 (Fig 7C), while root K+ accumulation remained unaltered by TIP2;2 overexpression (Fig 7D). No significant effect on ion accumulation was observed in the C24 background, independent of HKT1 expression (S11 Fig). These results indicate that AtTIP2;2 may have unique transport functions depending on the organ, and may enhance vacuolar storage of $Na^+$ in the root while promoting vacuolar $K^+$ retention in the shoot. The enhanced $Na^+$ and $K^+$ accumulation demonstrated here shows a direct requirement for $UAS_{GAL4}$:HKT1 and its dependency on genetic background.

## Discussion

HKT1 is widely recognized for its role in $Na^+$ retrieval from the xylem, reducing shoot $Na^+$ accumulation and enhancing salt tolerance in various plant species [1,3,13,16,52]. Our findings reinforce that HKT1 activity is also linked to root system architecture (RSA), particularly in regulating lateral root development under salt stress (Fig 8). While root architecture undergoes active reprogramming in response to salinity [9], a trade-off exists between $Na^+$ exclusion and lateral root development, which appears to be mediated by HKT1 [5,10]. This study sought to determine whether HKT1-mediated $Na^+$ transport influences RSA through transcriptional networks, particularly those involved in ABA signaling. By analyzing two independent UASGAL4:HKT1 overexpression lines in Col-0 (E2586) and C24 (J2731) that slightly differ in HKT1 expression domains, we aimed to disentangle the role of stele $Na^+$ accumulation from broader regulatory mechanisms. The transcriptomic responses in these two backgrounds revealed both conserved and genotype-specific expression patterns, emphasizing the complexity of salt stress adaptation.

Our results indicate that genetic background plays a critical role in shaping transcriptional responses to salt stress and HKT1 overexpression. While both Col-0 and C24 exhibit HKT1-dependent lateral root inhibition, their underlying gene expression patterns differ. TMAC2 emerged as a key candidate, as its regulation by salt and HKT1 expression was opposite in the two backgrounds. No major allelic differences were identified between Col-0 and C24 in the TMAC2 promoter region (S6 Fig). On the other hand, we observed differences when the same TMAC2 allele was overexpressed in both

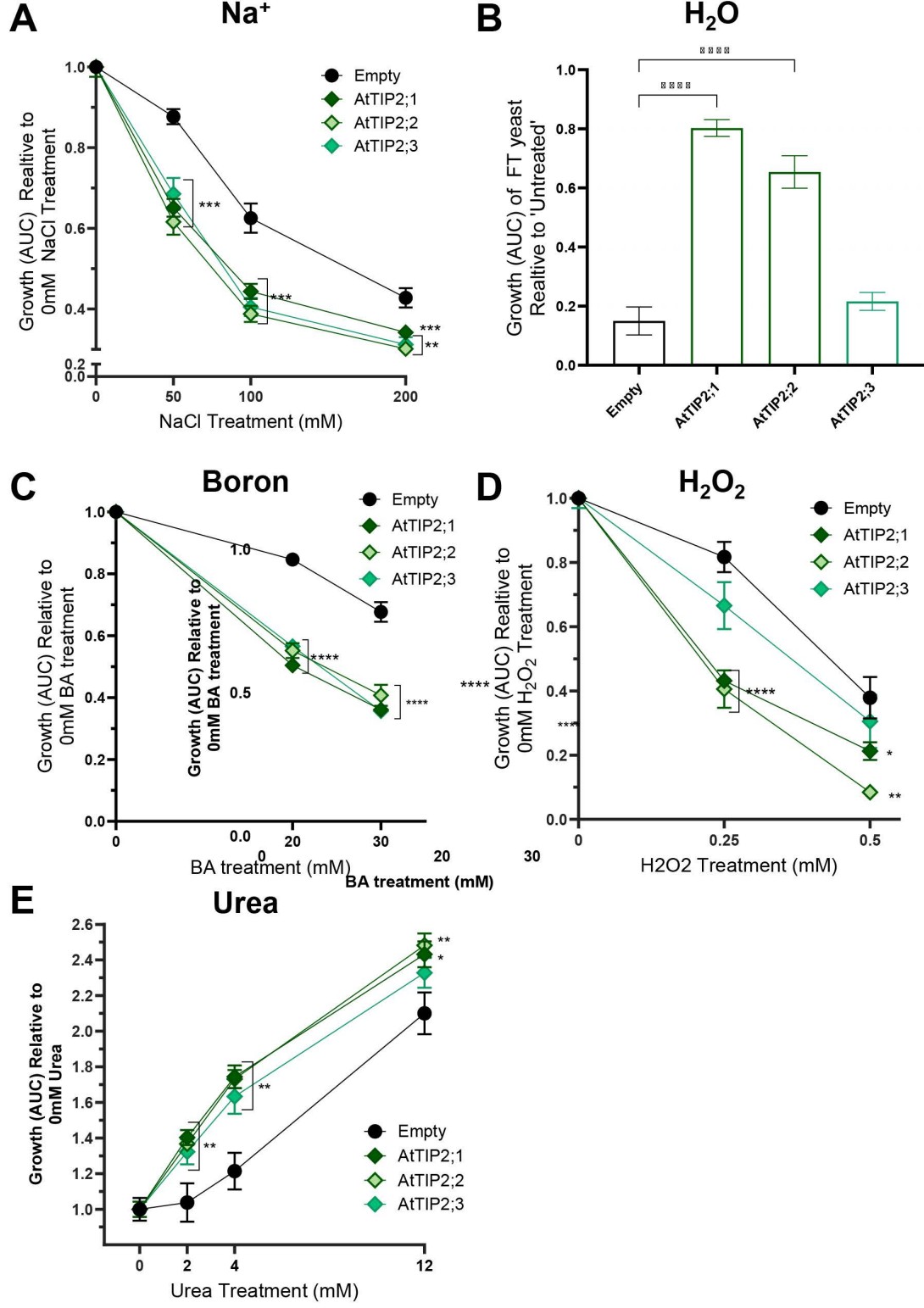

**Fig 6. Solute transport profile phenotypes for AtTIP2;2 and its closest homologs, AtTIP2;1 and TIP2;3.** High-throughput yeast screens were conducted to test for **(A)** Sodium, **(B)** Water, **(C)** Boron, **(D)** Hydrogen peroxide ($H_2O_2$) and **(E)** Urea permeability. Growth phenotypes were derived following [38] by calculating the Area Under the curve (AUC) from yeast $OD_{650}$ vs time graphs for each culture exposed to various treatments. Sodium,

Boron and $H_2O_2$ are toxicity-based screens, with decreased yeast growth/increased toxicity phenotypes of positive candidate TIP-expressing yeast cells, compared to an Empty vector control, indicative of TIP-mediated solute transport. Water transport assay (Freeze-thaw screen) attributes yeast survivorship enhancements of positive candidate TIP-containing yeast cells compared to Empty vector negative controls, reflective of respective levels of water-efflux/transport across membranes upon freezing and thawing treatments. The Urea growth-based assay enables comparison of growth enhancements of yeast grown in low-nitrogen treatments, where positive candidate TIP-expressing yeast displayed enhanced growth compared to Empty vector control, indicative of TIP-mediated urea transport. Asterisks denote One-Way ANOVA results comparing Treated yeast growth against Empty vector; *p<0.05, **p<0.01 and ***p<0.001. N=6, Error bars=SE. AtTIP2;1 and AtTIP2;2 were both identified as putative candidates for transport of sodium, water, boron, $H_2O_2$ and urea, resulting in enhancements in toxicity or growth/survivorship phenotypes in the respective assays. AtTIP2;3 was identified as a positive candidate for sodium, boron and urea transport, but not for water and $H_2O_2$.

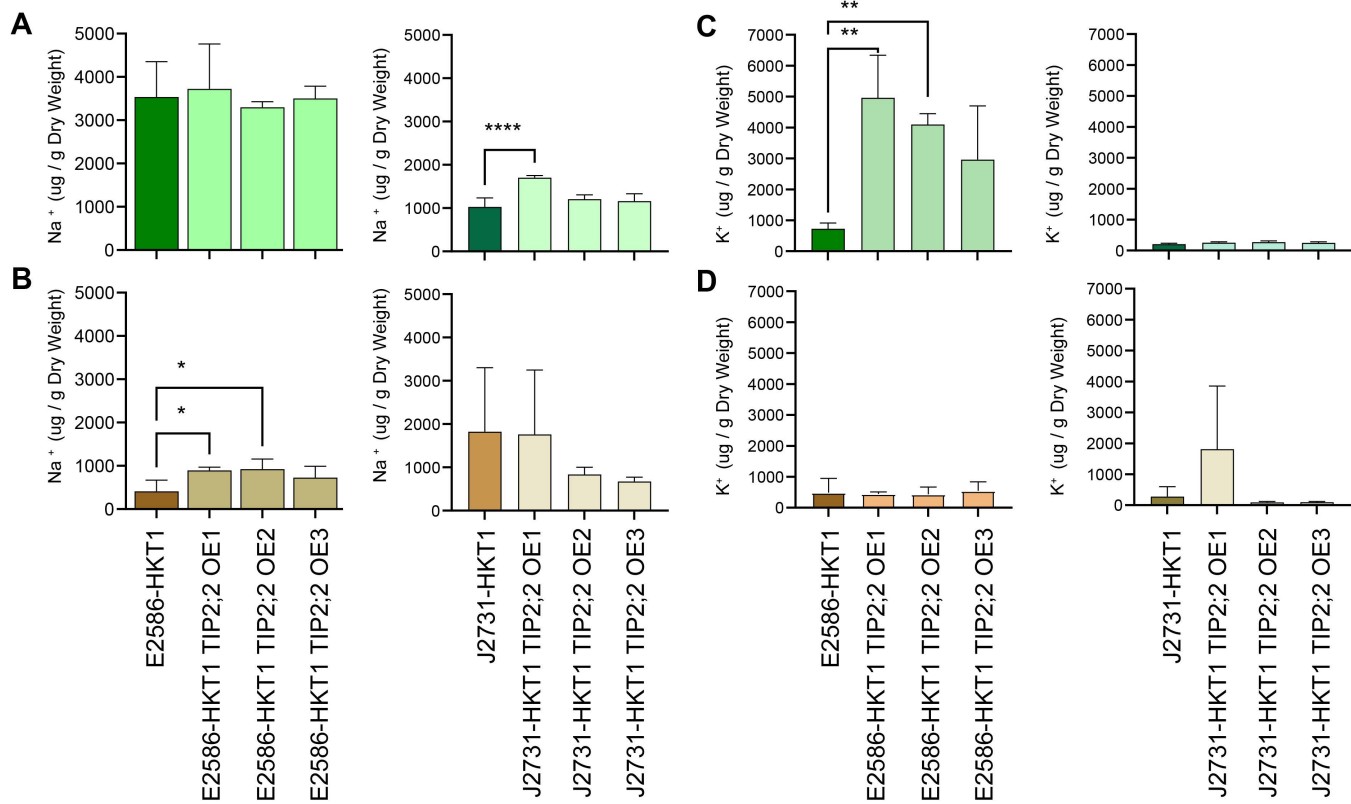

**Fig 7. TIP2;2 contributes to sodium compartmentalization in the root.** Four days old Arabidopsis seedlings from lines with tissue-specific HKT1 overexpression in Col-0 (E2586) and C24 (J2731) backgrounds with and without TIP2;2 overexpression were exposed to salt stress (75 mM NaCl) for 21 days. The sodium ($Na^+$) accumulation was measured in seedling's **(A)** shoot and **(B)** root tissue, as well as potassium ($K^+$) accumulation was measured in seedling's **(C)** shoot and **(D)** root. The bars represent the mean value calculated from at least 20 seedlings, and the error bars represent standard error. The significant differences between individual mutant lines and their respective background lines were determined using one-way ANOVA test, with *, **, *** and **** indicating p-values below 0.05, 0.01, 0.001 and 0.0001 respectively.

backgrounds, suggesting that the differences between Col-0 and C24 are due to downstream signals of TMAC2. In Col-0, TMAC2 overexpression resulted in increased ABA accumulation and lateral root inhibition, suggesting a role in enhancing ABA-mediated root growth suppression. In contrast, in C24, TMAC2 overexpression dampened ABA increase in response to HKT1 overexpression, pointing to a potential negative feedback mechanism that limits ABA accumulation. While in Col-0 we observed that this feedback loop is likely regulated through ABI4/5, transcription factors known to regulate ABA responses and $Na^+$ transport [53], the effect of TMAC2 on ABA in C24 involves another signaling node, that is thus far unexplored. Given that ABI4 has been shown to directly bind to the HKT1 promoter, it is plausible that this regulatory

 

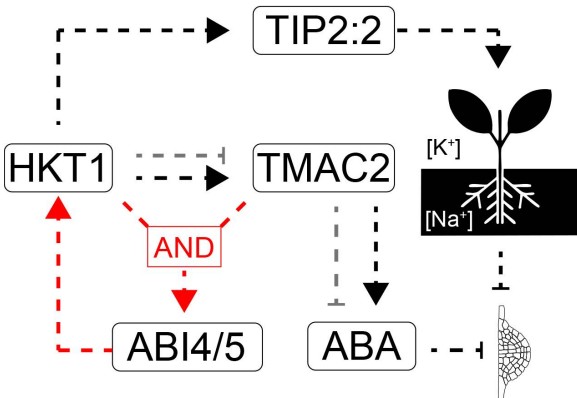

**Fig 8. Working model integrating TIP2:2 and TMAC into HKT1-based regulation of lateral root development.** The individual dashed lines represent the relationships between individual molecular components, with arrows and flat-head arrows indicating activating and repressing relationships respectively. The gray arrows indicate the alternative state of the relationship in C24. Red arrows indicate feedback loop. The pictograms illustrate potassium and sodium accumulation in whole shoot and root tissue respectively, and lateral root primordium development.

circuit links ABA signaling and Na⁺ transport in a dynamic feedback loop. Summarizing, HKT1 regulates lateral root development via an ABA-dependent feedback loop (Fig 8), albeit it is contingent on presence/absence of signaling components downstream of TMAC2.

Beyond the HKT1-TMAC2 interaction, TIP2;2 emerged as an additional regulator of lateral root inhibition under salt stress. TIP proteins contribute to water and neutral solute transport [54–57], and our findings suggest that TIP2;2 may also be involved in Na⁺ homeostasis. TIP2;2 was upregulated in response to both salt stress and HKT1 overexpression, and loss-of-function TIP2;2 mutants exhibited increased lateral root development, whereas overexpression lines displayed enhanced root inhibition. These findings, together with the subcellular localization of TIP2;2 in both plasma and tonoplast membranes (Fig 5), suggest that TIP2;2 may contribute to Na⁺ compartmentalization, indirectly impacting RSA under salt stress. While TIP2;2 expression was not enriched in lateral root primordia, its role in vacuolar Na⁺ sequestration may influence local osmotic balance and ion availability, thereby affecting root development. Given that TIP2;2 shares functional similarities with TIP2;1 (Fig 6), it is possible that other aquaporins contribute to lateral root inhibition under salinity, a possibility that warrants further investigation.

This study provides new insights into how HKT1, TMAC2, and TIP2;2 contribute to salt stress adaptation through distinct but interconnected pathways (Fig 8). HKT1-mediated Na⁺ sequestration reduces lateral root development via ABA signaling, with TMAC2 acting as a key modulator, involving ABI transcription factor feedback. Increased sodium accumulation in root stele increases expression of TIP2;2 that further contributes to Na⁺ compartmentalization, on both tissue- and subcellular level, further contributing to reduced lateral root development. Given the complexity of these interactions, future studies should focus on dissecting the genetic and molecular components that regulate HKT1-dependent root remodeling. Investigating ABA signaling components downstream of TMAC2 in both Col-0 and C24 could provide further insight into the genetic basis of stress adaptation. Likewise, elucidating the functional consequences of TIP2;2-mediated ion transport in lateral root development and expanding the analysis to additional aquaporins could help clarify their role in salt tolerance.

Taken together, this study sheds light on the molecular mechanisms linking Na⁺ transport, ABA signaling, and lateral root development under salt stress. By integrating transcriptomic analysis, mutant phenotyping, and gene expression studies, we propose a working model in which HKT1, TMAC2, and TIP2;2 collectively regulate root system architecture in response to salinity (**Fig 8**). Our findings highlight the importance of genetic background in shaping stress responses and

suggest that HKT1's role extends beyond Na⁺ homeostasis to influence broader developmental and hormonal networks. These results provide a foundation for future studies aimed at improving salt tolerance in crops by targeting key regulators of root system architecture.

## Supporting information

**S1 Fig. KEGG category enrichment for differentially expressed genes for bulk-comparisons.** The abundance of differentially expressed genes was evaluated per KEGG category for pair-wise comparisons **(A)** $UAS_{GAL4}$:*HKT1* versus background; **(B)** control 0 mM versus stress at 75 mM NaCl and **(C)** control 0 mM NaCl versus supplemental K+ (30 mM KCl) during salt stress (75 mM NaCl).
(PDF)

**S2 Fig.** Additional expression profiles of genes identified to have conserved expression in response to salt and HKT1. The genes shared between the genetic backgrounds and treatments were inspected for their expression relative to control conditions across treatments, with (A-B) two transcripts to show significant reduction in response to both NaCl and NaCl+KCl treatment, (C-F) four transcripts to show significant reduction in response to NaCl+KCl, (G-H) two transcripts to show significant increase in response to both NaCl and NaCl+KCl treatments, and (I-J) two transcripts to show increase in response to NaCl+KCl treatment. The p-values listed above the individual comparisons between UAS-HKT1 and their background were calculated using T-test.
(PDF)

**S3 Fig. Contributions of vacuolar phosphate transporter (AT1G63010 – PHT5:1) to salt-induced changes in root architecture.** Root System architecture of **(A)** T-DNA insertion lines (SALK_023873, SAIL_789_E03 and SALK_098188C) **(B)** Single and multiple mutant lines obtained from Dr. Chiou Lab, and **(C)** gain-of-function liens obtained from Dr. Chiou Lab (T.-Y. Liu et al. 2016) [58]. The 4 days old seedlings were exposed to control (0 mM NaCl) or salt stress treatment (75 or 125 mM NaCl) for 4 (control) and 12 days (salt stress) of treatment prior to quantification of root architecture using SmartRoot. The graphs represent individual components of root architecture: Main Root Length, lateral root number and average lateral root length. The significant differences between individual mutant lines and their respective background lines were determined using one-way ANOVA test, with *, **, *** and **** indicating p-values below 0.05, 0.01, 0.001 and 0.0001 respectively.
(PDF)

**S4 Fig. Contributions of UDP-dependent glycosyltransferase 76 B1 (AT3G11340 – UGT76B1) to salt-induced changes in root architecture.** Root System architecture of two overexpression (OX) lines and one loss-of-function line obtained from Dr. Schaffner Lab [59]. The 4 days old seedlings were exposed to control (0 mM NaCl) or salt stress treatment (75 or 125 mM NaCl) for 4 (control) and 12 days (salt stress) of treatment prior to quantification of root architecture using SmartRoot. The graphs represent individual components of root architecture: Main Root Length, lateral root number and average lateral root length. The significant differences between individual mutant lines and their respective background lines were determined using one-way ANOVA test, with *, **, *** and **** indicating p-values below 0.05, 0.01, 0.001 and 0.0001 respectively.
(PDF)

**S5 Fig. Contributions of CO2 responsive secreted protease (AT1G20160 - SBT5.2) to salt-induced changes in root architecture.** Root System architecture of T-DNA insertion lines (SALK_099861 and SALK_012112). The 4 days old seedlings were exposed to control (0 mM NaCl) or salt stress treatment (75 or 125 mM NaCl) for 4 (control) and 12 days (salt stress) of treatment prior to quantification of root architecture using SmartRoot. The graphs represent individual components of root architecture: Main Root Length, lateral root number and average lateral root length. The significant

differences between individual mutant lines and their respective background lines were determined using one-way ANOVA test, with *, **, *** and **** indicating p-values below 0.05, 0.01, 0.001 and 0.0001 respectively.
(PDF)

**S6 Fig. Comparison of genomic TMAC2 promoter and protein coding region between Col-0 and C24 accessions.** The sequences of Col-0 and C24 TMAC2 promoter regions have been retrieved from the SALK 1001 genomes browser (http://signal.salk.edu/atg1001/3.0/gebrowser.php) and aligned using JalView. The nucleotides that differ between Col-0 and C24 have been highlighted. The sequence coding the mRNA has been underlined with red.
(PDF)

**S7 Fig. Expression of TMAC2 in generated overexpression lines.** Expression of TMAC2 in Col-0 (E2586) and C24 (J2731) background with or without additional tissue-specific overexpression of HKT1. All expression results are based on three independent biological replicates collected from the leaves of soil grown plants. The significant differences between individual mutant lines and their respective background lines were determined using one-way ANOVA test, with *, **, *** and **** indicating p-values below 0.05, 0.01, 0.001 and 0.0001 respectively.
(PDF)

**S8 Fig. Shoot tissue ABA accumulation in lines overexpressing TMAC2.** The ABA accumulation in Col-0 (E2586) and C24 (J2731) seedlings with and without tissue specific overexpression of HKT1 and additional over-expression of TMAC2 was measured in shoots (green graphs) and roots (brown graphs) Arabidopsis seedlings 21 days after transfer to 0 or 75 mM NaCl.
(PDF)

**S9 Fig.** Sodium (Na+) accumulation in lines overexpressing TMAC2. Four days old Arabidopsis seedlings from lines with tissue-specific HKT1 overexpression in Col-0 (E2586) and C24 (J2731) backgrounds with and without TMAC2 overexpression were exposed to salt stress (75 mM NaCl) for 21 days. The sodium (Na+) accumulation was measured in seedling's shoot (green graphs) and root tissue (brown graphs). The bars represent the mean value calculated from at least 20 seedlings, and the error bars represent standard error. The significant differences between individual mutant lines and their respective background lines were determined using one-way ANOVA test, with *, **, *** and **** indicating p-values below 0.05, 0.01, 0.001 and 0.0001 respectively.
(PDF)

**S10 Fig. Expression of TIP2;2 in generated overexpression lines.** Expression of TIP2;2 in Col-0 (E2586) and C24 (J2731) background with or without additional tissue-specific overexpression of HKT1. All expressions are based on three independent biological replicates collected from the leaves of soil grown plants. The significant differences between individual mutant lines and their respective background lines were determined using one-way ANOVA test, with *, **, *** and **** indicating p-values below 0.05, 0.01, 0.001 and 0.0001 respectively.
(PDF)

**S11 Fig.** Sodium (Na+) and potassium (K+) accumulation under control (0 mM NaCl) and salt (75 mM NaCl) conditions of TIP2;2 overexpressing lines. Four days old Arabidopsis seedlings from lines with tissue-specific HKT1 overexpression in Col-0 (E2586) and C24 (J2731) backgrounds with and without TIP2;2 overexpression were exposed to salt stress (75 mM NaCl) for 21 days. The sodium (Na+) and potassium (K+) accumulation was measured in seedling's shoot (green graphs) and root tissue (brown graphs). The bars represent the mean value calculated from at least 20 seedlings, and the error bars represent standard error. The significant differences between individual mutant lines and their respective background lines were determined using one-way ANOVA test, with *, **, *** and **** indicating p-values below 0.05, 0.01, 0.001 and 0.0001 respectively.
(PNG)

**S12 Fig.** Sodium (Na+) and potassium (K+) accumulation under control and salt stress conditions in tip2;2 mutant lines. Four days old Arabidopsis seedlings from Col-0 and tip2;2 T-DNA insertion line (SALK_152463) were exposed to control (0 mM NaCl) or salt stress (75 mM NaCl) for 21 days. The sodium (Na+) accumulation was measured in seedling's shoot (green graphs) and root tissue (brown graphs) under (A) control or (B) salt stress conditions. Additionally, potassium (K+) accumulation was also measured in seedling's shoot (green graphs) and root tissue (brown graphs) under (C) control or (D) salt stress conditions. The bars represent the mean value calculated from at least 20 seedlings, and the error bars represent standard error. The significant differences between individual mutant lines and their respective background lines were determined using one-way ANOVA test, with *, **, *** and **** indicating p-values below 0.05, 0.01, 0.001 and 0.0001 respectively. (PDF)

**S13 Fig.** Representative example yeast growth curves, $Ln(OD/OD_i)$ vs. time, of yeast expressing Empty vector Negative control and AtTIP2;2. Respective yeast cultures were exposed to **(A)** 0mM, 50mM, 100MM and 200mM NaCl, for sodium toxicity assay; **(B)** Untreated and Freeze-thawed treated for Water transport assay; **(C)** 0mM, 20mM and 30mM Boric acid for Boron toxicity assay; **(D)** 0mM, 0.25mM and 0.5mM $H_2O_2$, for $H_2O_2$ toxicity assay and **(E)** 12mM, 4mM, 2mM and 0mM Urea for urea growth-based screen. Growth comparisons shown in Fig 7 were captured from the area under the curves (AUC) until the vertical dashed lines (measuring time point) show on $Ln(OD/OD_i)$ vs. time graphs. AtTIP2;2- expressing yeast displays reduced growth over time at each of the sodium, boric acid, H2O2 concentrations, indicative of an increased sensitivity/toxicity response to these treatments compared to Empty vector control. AtTIP2;2-expressing yeast also shows increased survivorship/recovery post freeze-thaw treatment, compared to Empty vector control which shows minimal growth post treatment exposure. AtTIP2;2-expressing yeast showed enhanced yeast growth over time at low nitrogen media concentrations (2mM and 4mM urea) compared to Empty vector control. (PDF)

**S1 Table.** List of mutants screened and primers used. (XLSX)

**S2 Table.** Primers used for cloning individual genes and links to their vector maps. (XLSX)

**S3 Table.** List of primers used for expression. (XLSX)

**S4 Table.** Relative expression of all mapped genes. (XLSX)

**S5 Table.** Average and Standard Error of gene expression. (XLSX)

**S1 Data.** Contains all of the raw data used for generating the figures within this manuscript. (XLSX)

## Acknowledgments

We would like to thank Dr. Chiou and Dr. Schaffner for sharing with us their genetic materials on PHT5:1 and UGT76B1 respectively.

## Author contributions

**Conceptualization:** Mark A. Tester, Magdalena M. Julkowska.

**Data curation:** Nouf O. Alshareef, Vanessa J. Melino, Annamaria De Rosa, Jian You Wang, Magdalena M. Julkowska.

**Formal analysis:** Nouf O. Alshareef, Vanessa J. Melino, Noha Saber, Annamaria De Rosa, Elodie Rey, Jian You Wang, Magdalena M. Julkowska.

**Funding acquisition:** Caitlin Byrt, Mark A. Tester.

**Investigation:** Nouf O. Alshareef, Vanessa J. Melino, Noha Saber, Annamaria De Rosa, Jian You Wang, Magdalena M. Julkowska.

**Methodology:** Noha Saber, Annamaria De Rosa, Elodie Rey, Jian You Wang, Magdalena M. Julkowska.

**Project administration:** Salim AlBabili, Caitlin Byrt, Magdalena M. Julkowska.

**Resources:** Salim AlBabili.

**Supervision:** Vanessa J. Melino, Salim AlBabili, Mark A. Tester, Magdalena M. Julkowska.

**Validation:** Nouf O. Alshareef, Vanessa J. Melino, Noha Saber, Annamaria De Rosa, Magdalena M. Julkowska.

**Visualization:** Nouf O. Alshareef, Vanessa J. Melino, Annamaria De Rosa, Elodie Rey, Magdalena M. Julkowska.

**Writing – original draft:** Nouf O. Alshareef, Vanessa J. Melino, Magdalena M. Julkowska.

**Writing – review & editing:** Noha Saber, Annamaria De Rosa, Elodie Rey, Jian You Wang, Salim AlBabili, Caitlin Byrt, Mark A. Tester.

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
