## [Decision Letter · Decision Letter 0]

PGENETICS-D-24-01275

Root Remodeling Mechanisms and Salt Tolerance Trade-Offs: The Roles of HKT1, TMAC2, and TIP2;2 in Arabidopsis

PLOS Genetics

Dear Dr. Julkowska,

Thank you for submitting your manuscript to PLOS Genetics. After careful consideration, we feel that it has merit but does not fully meet PLOS Genetics's publication criteria as it currently stands. Your study has been evaluated by three expert reviewers who, as you will see below, while acknowledging that the experiments are rigorous and the data convincing, have raised a number of significant concerns. We invite you to submit a revised version of the manuscript that addresses the points raised during the review process.

To improve the clarity and impact of manuscript, we recommend that you provide a clearer and more focused narrative throughout the paper. First, the central hypothesis of the study should be explicitly stated, particularly regarding how TMAC2, HKT1, and TIP2.2 interact to regulate lateral root development under salt stress. The connection between increased ABA sensitivity and Na+ accumulation in the pericycle needs further clarification, as the current text is ambiguous. Additionally, the potential role of HKT1 in lateral root inhibition requires a more critical and balanced discussion, especially in light of the mixed results between the two Arabidopsis ecotypes used in your study. We also suggest that the Introduction be reorganized to provide a clearer rationale for the use of these ecotypes, as well as a more thorough discussion of the genetic differences that may explain the varying results observed. In terms of the data, we encourage you to refine the transcriptomic analysis to better align with your research question and consider whether some of the less relevant data can be omitted or moved to supporting information. Finally, and most importantly, we urge you to address the concerns raised by Reviewer 1 regarding the overemphasis on HKT1 and the need for more comprehensive experimental design to substantiate the causal link between ABA signaling and lateral root development. Please note that we will only consider a revised manuscript for further evaluation if the concerns raised above are carefully addressed.

Please submit your revised manuscript within 60 days Mar 07 2025 11:59PM. If you will need more time than this to complete your revisions, please reply to this message or contact the journal office at plosgenetics@plos.org. Please include the following items when submitting your revised manuscript:

We look forward to receiving your revised manuscript.

Kind regards,

Paula Duque

Guest Editor

PLOS Genetics

Claudia Köhler

Section Editor

PLOS Genetics

Aimée Dudley

Editor-in-Chief

PLOS Genetics

Anne Goriely

Editor-in-Chief

PLOS Genetics

**Journal Requirements:**

At this stage, the following Authors/Authors require contributions: Nouf Alshareef, Vanessa Melino, Noha Saber, Annamaria De Rosa, Elodie Rey, Jian You Wang, Salim AlBabili, Caitlin Byrt, Mark Tester, and Magdalena M Julkowska. Please ensure that the full contributions of each author are acknowledged in the "Add/Edit/Remove Authors" section of our submission form.

The list of CRediT author contributions may be found here: https://journals.plos.org/plosgenetics/s/authorship#loc-author-contributions

https://journals.plos.org/plosgenetics/s/submission-guidelines#loc-parts-of-a-submission

4) We noticed that you used the phrase 'not shown' in the manuscript. We do not allow these references, as the PLOS data access policy requires that all data be either published with the manuscript or made available in a publicly accessible database. Please amend the supplementary material to include the referenced data or remove the references.

5) We do not publish any copyright or trademark symbols that usually accompany proprietary names, eg ©,  ®, or TM  (e.g. next to drug or reagent names). Therefore please remove all instances of trademark/copyright symbols throughout the text, including:

- TM on page: page 5 line 158.

6) Please upload all main figures as separate Figure files in .tif or .eps format. For more information about how to convert and format your figure files please see our guidelines: 

7) We have noticed that you have uploaded Supporting Information files, but you have not included a list of legends. Please add a full list of legends for your Supporting Information files after the references list.

8) We note that your Data Availability Statement is currently as follows: "All data is either attached within this manuscript or otherwise publically available". Please confirm at this time whether or not your submission contains all raw data required to replicate the results of your study. Authors must share the “minimal data set” for their submission. PLOS defines the minimal data set to consist of the data required to replicate all study findings reported in the article, as well as related metadata and methods (https://journals.plos.org/plosone/s/data-availability#loc-minimal-data-set-definition).

9) Please amend your detailed Financial Disclosure statement. This is published with the article. It must therefore be completed in full sentences and contain the exact wording you wish to be published.

1) Please clarify all sources of financial support for your study. List the grants, grant numbers, and organizations that funded your study, including funding received from your institution. Please note that suppliers of material support, including research materials, should be recognized in the Acknowledgements section rather than in the Financial Disclosure

2) State the initials, alongside each funding source, of each author to receive each grant. For example: "This work was supported by the National Institutes of Health (####### to AM; ###### to CJ) and the National Science Foundation (###### to AM)."

3) State what role the funders took in the study. If the funders had no role in your study, please state: "The funders had no role in study design, data collection and analysis, decision to publish, or preparation of the manuscript.".

If you did not receive any funding for this study, please simply state: u201cThe authors received no specific funding for this work.

10) Your current Financial Disclosure states, "The author(s) received no specific funding for this work.".

However, your funding information on the submission form indicates receiving a fund. Please ensure that the funders and grant numbers match between the Financial Disclosure field and the Funding Information tab in your submission form. Note that the funders must be provided in the same order in both places as well.

Please indicate by return email the full and correct funding information for your study and confirm the order in which funding contributions should appear. Please be sure to indicate whether the funders played any role in the study design, data collection and analysis, decision to publish, or preparation of the manuscript.

**Reviewers' comments:**

Reviewer's Responses to Questions

**Comments to the Authors:**

**Please note that one of the reviews is uploaded as an attachment.**

Reviewer #1: In this work, Alshareef et al investigated that role of TMAC2 (a negative transcriptional regulator of ABA) in lateral root development in two Arabidopsis ecotypes. The authors reported that overexpression of TMAC2 modulated root ABA content, affecting lateral root development, and that these changes were ecotype-specific. It is concluded that HKT1 and TMAC2 may form a positive feedback loop in ABA signalling operating upstream of lateral root development.

I must confess I have rather mixed feelings about this work. On one hand, the paper addresses a very important topic – plant adaptation to salinity stress, and the studies on the role of root system architecture in this process are relatively few and investigated much less that for other adaptive traits, such as Na exclusion or stomata patterning. On the other hand, the paper reads like a “dog’s breakfast” and mixes together various components without proving the existence of any casual relationship between them.

My first major issue is with an attempt to make HKT1 operation central to this story. With a due respect, this transporter is only one (of many) that determine tissue Na content. Thus, the entire working hypothesis per se looks rather vague to me. The authors then add two different ecotypes to the story, and report rather opposite effects of TMAC2 on lateral root development. While overexpression of TMAC2 in the Col-0 background increased ABA accumulation, its overexpression in C24 reduced ABA accumulation. Calling this a “genotype-specific effect” does not explain much and merely a way of confirming that the authors could not explain the reasons between these controversial results. The same is also true for the causal link between ABA and lateral root development. Many other factors (e.g., ethylene, ROS, etc) maya also contribute to this process. So, the entire story is outstretched, and conclusions are not substantiated by the data.

In the light of this, I believe the authors should be less bias in their assumption of HKT1 being central to lateral root development and provide more comprehensive and less subjective approach to investigate the causal link between ABA signalling and RSA formation. This implies omitting some (not directly relevant) data and conducting additional experimentation.

Minor issues:

- Ln 28. Describe TMAC2 abbreviation.

- Ln 50-55. This is a rather eclectic list, mixing together various traits and processes, without any structural logic.

- Ln 79 and elsewhere. “Johannes D. Scharwies et al 2024”. Why using first names in references?! This looks like a student’s style of writing to me. Have any of co-authors read and approved the MS? …

Reviewer #2: This manuscript describes some interesting new results regarding the role of TMAC2 (a negative ABA-regulator) and of a tonoplast aquaporin-type transporter (TIP2.2) in lateral root inhibition under salt stress. The data are interesting and new. However, I struggled with some of the text especially in the Introduction and the early part of the Results. I did not completely grasp the precise question addressed (role of HKT1?) and certainly lost sight of it early on. My comments below just concentrate on these aspects, but the whole paper should be edited to follow a clear line of hypothesis testing.

The Introduction needs a better description of the expression patterns of the overexpressed HKT1 in J2731 and E2586 and of the corresponding phenotypes. If I understand correctly from the previous publications both lines overexpress HKT1 in the stele but overexpression includes/excludes the pericycle in J2731 and E2586. (This is repeated in this manuscript but legend of Fig. 6 in Julkowska et al (Plant Cell, 2017) notes “UAS-HKT1 lines, with enhanced HKT1 expression in the root pericycle”, I assume this is a mistake?).

Moller et al reported a decrease in shoot Na for both overexpression lines but the effect is stronger in J2731; UASGAL4:HKT1 than in E2586; UASGAL4:HKT (mentioned in the Introduction of this manuscript). The Plant Cell Julkowska et al 2017 paper (Fig. 6) showed inhibition of LR development/growth in both overexpression lines but this phenotype is stronger in E2586; UASGAL4:HKT than in J2731; UASGAL4:HKT1 (not mentioned in the Instroductio of this manuscript). Is this correct?

If so, could the authors please justify the hypothesis that the effect on lateral root development is due to Na accumulation in the pericycle. Would the latter not be expected to be stronger in J2731; UASGAL4:HKT1 than in E2586; UASGAL4:HKT.?

The precise research question (how does HKT1 alter LR development?) should be explicit, and then followed through in the Results and Discussion.

Results:

I find the initial description of the transcriptome data distracts from the research question. For example, enrichment of annotations for some pairwise comparisons shown in Fig 1 seems to be only of tangential interest for this paper. Given the multi-factorial design of the experiment a more sophisticated analysis of the dataset would be possible and could be published separately. Here, Fig. 1B-C and their description could be omitted or moved to Supplemental material. I also find some of the conclusions unjustified. E.g. that a stronger transcriptional response in the Na+K treated plants compared to Na alone supports the observation that addition of K alleviates that root phenotype. One could just as well expect a weaker response given that the plants are less stressed.

From line 335 onwards there is more focus on the research question. To ‘identify the genes downstream of UAS-HKT1 that may be involved in the repression of lateral root development under salt stress’ the authors extract genes that show interesting differences in the response profiles between the lines. This makes sense. However, I do not understand the term ’similar’ in the legend of Fig. 2 (clearly the response patterns differ). More importantly, I find it difficult to match the profiles with the reported root phenotypes (see my comment on Introduction). Naiively, I would have looked for genes that show a similar difference to WT in both over-expression lines (as they are both inhibited in LR development) but with a stronger difference n E2586 than in J2731 background (see Fig. 6 in Julkowska et al 2017)) . Some genes show such pattern, others show a stronger response in J2731 (matching the Na removal phenotype reported in Moller) while TMAC actually has an opposite profile in J2731 and E2586. Why did the authors focus on this gene? It would be good to sort genes into different profiles and then systematically compare them with the reported phenotypes.

The Discussion could be better focussed. Hypothetical statements, especially not linked to the genes investigated should be omitted.

I struggle to understand what exactly is being concluded. Yes, clearly gene functions and responses differ in the two genetic backgrounds, which is probably not too surprising but what does this mean? Do the transcriptomics results help to pin down the differences? How many SNPs between Col-0 and C24???? Could it be that the previously reported different root phenotypes in the HKT1 overexpression lines are not due to differences in HKT1 localisation but to other differences in the two backgrounds?

The authors try to explain their observations with ‘feedback loops’ and ‘complex regulatory networks’. A scheme depicting at least a working hypothesis would certainly help the reader.

Reviewer #3: This is a very competent study of genes associated with the reduction of lateral root growth by salt stress. In particular it looks at the role of HKT1 or TIP2;2 overexpression lines in modifying the response patterns. As one would expect, the responses are complex. The responses are described well, but I could not find a hypothesis or the outline of a coherent narrative that the results would (or not) ‘fit’. This made the significance of the work difficult for me to fully understand.

The focus of the paper is not clear. Is the central topic the mechanism of reduction of lateral roots under salt stress? And to what extent this inhibition is explained by increased sensitively to ABA due to expression of TMAC2 and/or OX of HKT1? If so, this comparatively straightforward simple aim is not stated clearly. There seems to be ambivalence as to whether inhibition of lateral root growth is caused by increased ABA (line 97) or by increased Na+ in the pericycle (line 105).

How this all gels with function of TIP2;2 is not clear. A simple diagram of the connections between root growth mechanisms and salt tolerance is needed.

Lastly, I suggest that there should be information presented at the start of the paper, on the reason to compare the Arabidopsis ecotypes Col-0 and C24, and how they might differ in Ca signaling pathways or in whatever might explain a different result of the overexpression of HKT1. The paper should include a discussion of the results in relation to the genomic or transcriptomic differences of the two ecotypes. Just saying “genetic background” is not sufficient.

**Have all data underlying the figures and results presented in the manuscript been provided?**

Reviewer #1: Yes

Reviewer #2: Yes

Reviewer #3: Yes

PLOS authors have the option to publish the peer review history of their article (what does this mean? ). If published, this will include your full peer review and any attached files.

**Do you want your identity to be public for this peer review?** For information about this choice, including consent withdrawal, please see our Privacy Policy .

Reviewer #1: No

Reviewer #2: No

Reviewer #3: No

**Figure resubmission:**
---

## [Decision Letter · Decision Letter 1]

Dear Dr Julkowska,

We are pleased to inform you that your manuscript entitled "Root Remodeling Mechanisms and Salt Tolerance Trade-Offs: The Roles of HKT1, TMAC2, and TIP2;2 in Arabidopsis" has been editorially accepted for publication in PLOS Genetics. Congratulations!

Yours sincerely,

Paula Duque

Guest Editor

PLOS Genetics

Aimée Dudley

Editor-in-Chief

PLOS Genetics

Aimée Dudley

Editor-in-Chief

PLOS Genetics

Anne Goriely

Editor-in-Chief

PLOS Genetics

Comments from the reviewers (if applicable):

Reviewer's Responses to Questions

**Comments to the Authors:**

Reviewer #2: This is a much improved presentation of the work. I wish the authors luck with continuing this line of research in the future.

Reviewer #3: The authors have addressed all my comments to my satisfaction, and I consider the paper acceptable for publication.

**Have all data underlying the figures and results presented in the manuscript been provided?**

Reviewer #2: Yes

Reviewer #3: None

PLOS authors have the option to publish the peer review history of their article (what does this mean? ). If published, this will include your full peer review and any attached files.

**Do you want your identity to be public for this peer review?** For information about this choice, including consent withdrawal, please see our Privacy Policy .

Reviewer #2: **Yes: ** Anna Amtmann

Reviewer #3: No

**Data Deposition**

http://datadryad.org/submit?journalID=pgenetics&manu=PGENETICS-D-24-01275R1

**Press Queries**

---

## [Editor Report · Acceptance letter]

PGENETICS-D-24-01275R1

Root Remodeling Mechanisms and Salt Tolerance Trade-Offs: The Roles of HKT1, TMAC2, and TIP2;2 in Arabidopsis

Dear Dr Julkowska,

We are pleased to inform you that your manuscript entitled "Root Remodeling Mechanisms and Salt Tolerance Trade-Offs: The Roles of HKT1, TMAC2, and TIP2;2 in Arabidopsis" has been formally accepted for publication in PLOS Genetics! Your manuscript is now with our production department and you will be notified of the publication date in due course.

With kind regards,

Anita Estes

PLOS Genetics

On behalf of:
